# Identification of a Promising Novel Genetic Source for Rice Root-Knot Nematode Resistance through Markers Associated with Trait-Specific Quantitative Trait Loci

**DOI:** 10.3390/plants13162271

**Published:** 2024-08-15

**Authors:** Pallavi Mohanapure, Meghraj Chavhan, Divya Singh, Jyoti Yadav, Vishal Singh Somvanshi, S. Gopala Krishnan, K. K. Vinod, Prolay K. Bhowmick, Haritha Bollinedi, Ashok Kumar Singh, Uma Rao, Ranjith Kumar Ellur

**Affiliations:** 1Division of Genetics, ICAR-Indian Agricultural Research Institute, New Delhi 110012, India; prema.sankala@gmail.com (P.); pallavi25mailbox@gmail.com (P.M.); chavhanmeghraj@gmail.com (M.C.); gopal_icar@yahoo.com (S.G.K.); kkvinodh@gmail.com (K.K.V.); prolaybhowmick@gmail.com (P.K.B.); haritha.agrico@gmail.com (H.B.); aks_gene@yahoo.com (A.K.S.); 2Division of Nematology, ICAR-Indian Agricultural Research Institute, New Delhi 110012, India; divyasingh73@gmail.com (D.S.); jyotirbs007@gmail.com (J.Y.); vishal.somvanshi@gmail.com (V.S.S.); umanema@gmail.com (U.R.); 3Eliciton Innovations Pvt Ltd., Hyderabad 500001, India

**Keywords:** rice root-knot nematode resistance, SSR, F_2:3_ population, quantitative trait loci

## Abstract

Direct-seeded rice (DSR) is gaining popularity among farmers due to its environmentally safe and resource-efficient production system. However, managing the rice root-knot nematode (RRKN), *Meloidogyne graminicola*, remains a major challenge in DSR cultivation. Developing genetic resistance is a pragmatic and effective approach compared to using hazardous pesticides. Pusa Basmati 1121 (PB1121) is the most popular Basmati rice variety, but it is highly susceptible to RRKN. In contrast, Phule Radha (PR) has shown highly resistant reaction to RRKN, as reported in our earlier study. We generated an F_2:3_ population from the cross of PB1121/PR and evaluated it for RRKN resistance-related traits under artificial inoculation conditions. The distribution pattern of traits in the F_2:3_ population indicated that resistance may be governed by a few major-effect genes and many minor-effect genes. The molecular markers reported to be associated with QTLs governing RRKN resistance traits were used to test in the current population. Although the simple linear regression identified significant associations between the markers and RRKN resistance-associated traits, these associations were spurious as the LOD score was below the threshold limit. This indicates that PR possesses novel genomic regions for resistance to RRKN as it does not possess any of the earlier reported QTLs.

## 1. Introduction

Rice (*Oryza sativa* L.) is one of the most important staple food crops that feed over four billion people worldwide [1]. Nematodes are serious pests which cause annual losses of up to US 80 billion globally across crop species [2]. It has been reported that rice is infested by about 300 nematode species belonging to 35 genera [3]. Based on distribution, spectrum of infestation and economic importance, global nematological societies have ranked *Meloidogyne* spp. (root-knot nematode) as the top genus of plant-parasitic nematodes [3]. RRKN (*Meloidogyne graminicola*) is a serious pest across all the rice-growing regions, including Asia and south-east Asia, where 90% of rice is produced and consumed [4]. In India, the rice yield losses due to RRKN under well-drained soils have been reported to be up to 21% [5], while those under upland conditions ranged from 16–32% [6]. RRKN infestation has been recorded in nurseries as well as transplanted rice fields across India, including in the states of Punjab, Haryana, Himachal Pradesh, Uttar Pradesh, Delhi and Jammu [7].

RRKN is an obligate sedentary endoparasite with second-stage juveniles (J2s) as the sole infective stage. J2s can penetrate roots through the elongation zone, root cap and injured root tips. Within a day of penetration, J2s penetrate the root cortex and orient their body parallel to the stele. Further, they travel through the cortical cell layers towards the apical meristem, where they begin feeding. After 42 h of invasion, J2s are found in the apical meristem area. Within 72 h of inoculation, hyperplasia and hypertrophy of root cortical cells referred to as giant cells occur, resulting in root-knots or galls around the nematode establishment site. Giant cells grow in the stellar region of the nearest metaxylem. The mechanical disruption caused by galls in metaxylem arteries interrupts water and nutrient intake, severely impairing root physiology and development [8]. This results in nutritional deficiency, leaf chlorosis, wilting, loss of vigour, reduced crop growth and yield losses [4]. 

Although continuous flooding reduces the RRKN population, it is not a pragmatic approach due to scarcity of water brought about by global climatic vagaries, leading to uncertain rainfall patterns. Additionally, most of the nematicides used earlier are currently discontinued due to their harmful effects, and some new molecules like Valem Prime and Fluensulfone have a restricted label claim. Other alternative management approaches are not adequate. Therefore, exploring genetic resistance to RRKN is an effective, economical and environmentally sustainable strategy [9]. 

Most of the rice genotypes grown across all the rice cropping systems, including lowland, upland, deep-water and irrigated systems, are susceptible to RRKN [10]. However, some accessions of *Oryza glaberrima*, *Oryza longistaminata* [11] and *Oryza glumaepatula* [12] were found to be resistant to RRKN. The rice genotypes such as Abhishek, Suraksha, Vandana, EK70, Khalibagh, Phule Radha (PR), Zhonghua 11 and LD24 were also identified as resistant [7,10,13,14,15,16]. In Zhonghua 11, a hypersensitivity-like reaction with necrotic cells surrounding the nematodes was responsible for the incompatible interaction, leading to resistance [17]. A major dominant gene *MG1* located on the long arm of chromosome 11 encoding a coiled-coil, nucleotide-binding and leucine-rich repeat (CC-NB-LRR) protein was found to govern resistance to RRKN in the rice genotypes SL 22-620, HKG 98 and Zhonghua 11. The LRR domain is critical for the identification of nematode invasion and further activation of MG1-triggered defence response and cell death upon a series of interactions [18]. 

In a RIL population generated by crossing Annapurna (susceptible) and Ramakrishna (resistant), a major QTL, *qMg-3a*, with a phenotypic variance of 41.10% was demonstrated to be associated with nematode egg production [19]. Likewise, another major QTL was identified on chromosome 11 using two F_2_ populations derived from LD24/Vialone Nano and Khao Pathak Maw/ Vialone Nano [16]. 

A consistent QTL was identified on chromosome 6 as governing gall number at 2 and 4 weeks of inoculation, which explained 9.0% and 9.6% of the phenotypic variance, respectively, in the resistant rice genotype Bala [20]. Similarly, in a RIL population generated from a cross of IR78877-208-8-1-2 (resistant) and Dinarado (susceptible), the QTLs *qJ2RS2.1*, *qJ2GRT2.1*, *qJ2RS3.1* and *qJ2GRT3.1* associated with reduced nematode reproduction and two QTLs *qGR3.1* and *qGR5.1* associated with decreased root galling were mapped [21]. QTLs linked to resistance against RRKN have been identified in the populations from the cross between the aerobic rice genotype 208-B-1-2 and the *O. glaberrima* genotype CG14 [21,22,23]. 

Genome-wide association studies have identified several marker-trait associations (MTAs) associated with traits related to nematode resistance. In a rice diversity panel, 11 MTAs were identified for gall number [24] and 16 putative candidate genes underlying these MTAs were identified to be differentially expressed [25]. In a panel of wild rice accessions comprising of *O. nivara*, *O. rufipogon* and weedy rice accessions, the MTAs for gall number, egg mass, eggs per egg mass and multiplication factor were identified on chromosomes 1, 2, 3, 6, 10 and 11 [7]. However, exploitation of wild rice genotypes in a resistance breeding program is resource-intensive and poses several challenges. 

We had reported RRKN resistance in PR, which is an early-duration, semi-dwarf, high-yielding, short and slender fine-grain variety [26]. Further, our team had also revealed that the resistance in PR may be attributed to the high suberin deposition in the root exodermis, which acts as a penetrative barrier for RRKN [27]. However, the genomic regions governing RRKN resistance in PR are unknown. Therefore, as an initial step towards QTL mapping, the current study was conducted to examine the existence of any previously identified QTLs for RRKN resistance. 

## 2. Results

### 2.1. Characterization for RRKN Resistance-Related Traits

The one-way analysis of variance (ANOVA) yielded statistically significant differences (*p* < 0.01) (Table 1) among the parental lines for the traits associated with resistance to RRKN (Figure 1, Figure 2 and Figure 3 and Table 1). PB1121 showed highly susceptible reaction with a gall number (NG) of 5.36 ± 0.28 (Mean ± SE), females per plant (Fm) of 14.32 ± 0.50, egg masses per plant (EM) of 13.07 ± 0.51, eggs per egg mass per plant (EEM) of 81.7 ± 8, and multiplication factor (MF) of 35.97 ± 4.06. However, the genotype PR showed highly resistant reaction with an NG of 1 ± 0.17, F of 0.83 ± 0.17, EM of 0.43 ± 0.11, EEM of 10 ± 2.91, and MF of 0.4 ± 0.12.

The ANOVA revealed statistically significant differences for traits associated with RRKN resistance in the F_2:3_ population at *p* < 0.01 (Table 2). This illustrated the existence of substantial genetic variability for all the RRKN resistance-associated traits in the population. The descriptive statistics of the F_2:3_ population for RRKN resistance-related traits are presented in Table 3.

The population showed a nearly normal distribution for the traits NG, Fm and EEM with a moderate positive skewness; however, the traits EM and MF were highly positively skewed. Transgressive segregants on both sides were observed for the traits EM, EEM and MF; however, no transgressive segregants were identified for the trait Fm. For the trait NG, transgressive segregants were observed to exhibit a lower trait value (Figure 4).

### 2.2. Identification of Markers Polymorphic between Parental Lines

The SSR markers located in the genomic regions associated with the earlier reported QTLs were identified, and a polymorphism survey was conducted between the parental lines PB1121 and PR. Out of the 155 selected SSR markers, 31 were identified to be polymorphic between PB1121 and PR. The highest number of polymorphic markers were identified on chromosomes 1 and 2 with five markers each, followed by chromosomes 3, 11 and 12, each having four polymorphic markers. The chromosomes 4, 6, 9 and 10 possessed one, three, three and two polymorphic SSR markers, respectively, while the chromosomes 5, 7 and 8 possessed no polymorphic markers (Figure 5).

### 2.3. Single-Marker Analysis

All 31 polymorphic markers were used for genotyping the F_2:3_ population. However, two markers RM572 (*p* = 0.01784) and RM1284 (*p* = 0.033) showed segregation distortion. Therefore, the remaining 29 markers were used for further analysis. The simple linear regression analysis revealed a significant association of nine markers with the RRKN resistance-related traits. The marker RM24 was associated with the trait NG, which explained 3.8% of the phenotypic variance (R^2^). The marker RM5122 was associated with four RRKN resistance-related traits, viz., EEM (R^2^ = 5.0%), MF (R^2^ = 5.0%), EM (R^2^ = 3.0%) and F (R^2^ = 3.0%). RM552 was associated with three traits, viz., F (R^2^ = 4.0%), EEM (R^2^ = 5.0%), and MF (R^2^ = 4.0%). The marker RM6737 was linked to the traits EM (R^2^ = 3.0%) and MF (R^2^ = 3.0%). The marker RM23911 was associated with the traits EEM (R^2^ = 4.0%) and MF (R^2^ = 3.0%). The markers RM15652, RM20158, RM6293 and RM3713 were associated with the traits EEM (R^2^ = 3.0%), Fm (R^2^ = 4.0%), EM (R^2^ = 3.0%) and MF (R^2^ = 3.0%), respectively, at *p* < 0.05 (Table 4). However, the single-marker analysis revealed that the LOD values for these associations were very low, indicating that these associations were not robust (Appendix A).

## 3. Discussion

Rice crop is traditionally grown under puddled transplanted conditions, which involves raising seedlings in a nursery and transplantation of 21-day-old seedlings into the main field. This approach is water-intensive, time-consuming and expensive due to the resources involved in raising nurseries, uprooting and main field preparation through puddling and transplantation. The limited availability of labour during the peak transplanting season, uncertain irrigation water supply, increased groundwater depletion, increased greenhouse gas (methane) emissions and negative effects on soil physical and chemical properties make this technology difficult to sustain long-term crop yields. Transition from traditional transplanted rice to direct-seeded rice (DSR) cultivation offers significant advantages, especially by reducing water and labour requirement [28]. However, this shift is accompanied by emerging risks, including poor crop stand due to weed infestation, intensified soil-borne pathogens like nematodes, and challenges in nutrient management. Among these risks, RRKN infestation stands out as a critical concern. RRKN induces galls in rice roots, leading to a substantial reduction in root and shoot development. This damage is particularly pronounced under aerobic, zero-tilled and dry DSR conditions [29]. To address this threat, the utilization of genetic resistance emerges as the most significant and effective management approach [9]. The resistance to RRKN was identified in *O. glaberrima* and *O. longistaminata* [11], but most of the Asian rice cultivars are susceptible to RRKN except for a few cultivars such as PR. The popular Basmati rice variety PB1121 is highly susceptible to RRKN, while the genotype PR shows highly resistant reaction. The tremendous genetic divergence for all the RRKN resistance-related traits (NG, Fm, EM, EEM and MF) makes them ideal parental lines for the development of a mapping population to map the genes/QTLs governing RRKN resistance.

PF-127 is a transparent gel that helps in the stringent screening of genotypes for RRKN resistance [13]. It provides less opportunity for escapes as nematodes are inoculated around the root tip. Further, it is a convenient and simple approach to screen rice genotypes at a specific temperature and relative humidity as the screening could be carried out in Petri plates placed in an incubator under controlled conditions. This increases precision and reproducibility and enables high-throughput screening in small space for evaluation against RRKN. Additionally, it facilitates the scoring of galls and various other disease parameters in the entire root system, thereby avoiding errors during disease scoring. Owing to these advantages, the F_2:3_ population was screened for resistance to RRKN in PF-127 gel.

Earlier reports indicated that resistance to RRKN is predominantly controlled by both major- and minor-effect QTLs [7,16,20,23,24]. In the current study, the traits NG, Fm, EM and EEM possessed a normal distribution with slight positive skewness, while MF showed strong positive skewness. These observations demonstrate that these traits are complex in nature and governed by few major-effect and many minor-effect genes [23].

The analysis isolated families that were equivalent to or more susceptible than the susceptible parental PB1121 and families equivalent to or more resistant than the resistant parent PR. This indicates the presence of transgressive segregants and further strengthens the hypothesis that RRKN resistance in PR is governed by both major- and minor-effect genes. This also clearly suggests that PR is a promising donor source for developing RRKN resistance in rice varieties suited for DSR conditions.

With the advances in molecular markers, many major genes and QTLs have been identified that show resistance to nematodes in various crops. The major gene *Mi-1.2* in the wild tomato species *Lycopersicon peruvianum* [30], the gene *Ma* in *Myrobalan plum* spices [31] and, recently, *M. graminicola-resistance gene 1* (*MG1*) in rice [18] have been functionally characterized to govern nematode resistance. Two major QTLs for RRKN resistance were identified on chromosomes 11 and 6 [9,16]. Several minor-effect QTLs governing RRKN resistance-associated traits have also been identified. In total, 27 QTLs governing NG [9,20,21,23], 3 QTLs governing the number of eggs/plant [19] and 12 QTLs governing J2s per root system and J2s per gram of root weight [23] have been reported earlier. Additionally, 33 MTAs for NG, EM, EEM, MF and nematode resistance have been mapped across studies [7,24]. Furthermore, several QTLs associated with yield and yield-related traits under RRKN infestation have been reported [23].

We attempted to validate the earlier reported QTLs using the linked molecular markers in the F_2:3_ population generated from the cross of PB1121/PR. Although the simple regression analysis identified a significant association between the marker RM24 located on chromosome 1 with the trait NG, the LOD value was quite low (LOD = 1.6). Furthermore, it explained 3.8% of the phenotypic variance, which was low compared to the earlier reported PVE of 8.3%. Similarly, the remaining eight markers were also found to be associated with the RRKN resistance-related traits (*p* < 0.05) (Table 3). However, the single-marker analysis revealed LOD values that were way below the threshold. Therefore, it was concluded that the associations identified using simple linear regression were spurious, as demonstrated by the low LOD values when analysed using the single-marker analysis approach. Additionally, the genetic variation for RRKN resistance-related traits between PB1121 and PR is tremendous, which cannot be explained by these few minor-effect associations. This unequivocally demonstrates that PR could be a novel source of RRKN resistance, thereby warranting QTL mapping using high-density markers to identify novel genomic regions governing RRKN resistance.

## 4. Materials and Methods

### 4.1. Plant Materials

A cross was generated between the genotypes PB1121 and PR during the wet season of 2018 to generate the F_2:3_ population. PB1121 is a popular Basmati rice variety developed at ICAR-IARI, New Delhi, which is highly susceptible to RRKN. PR is an early-duration, semi-dwarf, high-yielding, short and slender fine-quality grain variety, developed from the cross of T(N)-1 × Kolamba 540 [32], which possesses high resistance to RRKN [26].

### 4.2. RRKN Culture and In Vitro Screening

A pure culture of RRKN that was regularly maintained on the susceptible rice variety PB1121 was used for in vitro screening during the wet season of 2021. J2s were extracted from mature root galls by the modified Baermann funnel technique and used for screening the rice parental lines and F_2:3_ population. The Pluronic F-127 gel has been established as a stringent medium for in vitro assays involving RRKN and rice [13]. Therefore, 23% Pluronic gel PF-127 medium was prepared, as reported earlier [13]. Seeds of the PR, PB1121 and F_2:3_ lines were surface-sterilized with 70% ethanol for 30 s to 1 min, followed by rinsing with sterile water 3 to 4 times and soaking overnight in sterilized distilled water. Sterilized rice seeds were placed on germination paper inside a Petri dish and incubated at 28 °C in a growth chamber. After 3 to 4 days of germination, seven seeds were placed on each plate containing 23% PF-127 gel. An average of 30 J2s of RRKN were inoculated at the root tip of each plant. Three plates of each genotype were included in each experiment. The plates were incubated at 28–30 °C with 16:8 h of light/dark photoperiod in a plant growth chamber (Labtech, France) maintained at the aforesaid condition for 15 days. The roots were harvested and stained with the NaOCl–Acid fuchsin method [33]. The roots were dissected under the microscope and observations on NG, Fm, EM and EEM were recorded on each plant. Photographs were captured using a Zeiss Axiocam MRm microscope. Nematode multiplication factor is estimated using the following formula:MF = EM × EEM/nematode inoculum level.

### 4.3. Molecular Marker Analysis

The literature survey identified a total of 82 QTLs associated with traits related to RRKN resistance (Appendix A). Notably, all chromosomes were reported to harbour QTLs linked to RRKN resistance. Chromosome 3 carried the highest number of QTLs (15), followed by chromosome 4 (13 QTLs) and chromosome 1 (9 QTLs). The remaining chromosomes carried QTLs ranging from 1 to 7.

Chromosome 1 possessed two QTL hotspots, one on the short arm represented by two QTLs and the other on the long arm represented by three QTLs. Chromosome 2 possessed one QTL hotspot represented by two QTLs. Chromosome 3 possessed two hotspots, wherein the first hotspot was represented by 6 QTLs and the other hotspot represented by 3 QTLs. Chromosome 4 possessed two hotspots represented by 5 and 3 QTLs. Chromosome 5 possessed one QTL hotspot and chromosome 6 had two hotspots represented by two QTLs each. Chromosome 12 possessed a hot spot with 4 QTLs (Figure 4).

Earlier studies have utilized SSR, SNP, RFLP and STS markers for linkage map construction and QTL mapping using biparental mapping populations with the population size ranging from 69 BC_1_F_1_ individuals [9] to 300 RILs [21]. Additionally, GWAS has been explored to identify MTAs. The phenotyping procedure adopted was either field-based or pot-based experiments, with a focus on the number of galls per root system. Few of the studies have also identified QTLs associated with eggs per root system, weight of roots after nematode infection and agronomic traits under RRKN infestation.

The linked SSR, SNP and RFLP markers were collated, and around 155 SSR markers were selected for polymorphic survey between PB1121 and PR.

During the wet season of 2021, the total genomic DNA was extracted using the technique demonstrated by Murry and Thompson [34], and quantification was carried out using a 0.8 percent agarose gel. The standard PCR mix containing dNTPs, Tris-HCl buffer, MgCl_2_, KCl and Taq-DNA polymerase was used, along with the primers and genomic DNA for carrying out PCR. The PCR program consisted of an initial denaturation for 5 min at 95 °C, followed by 35 cycles consisting of denaturation at 95 °C for 30 s, annealing at 55 °C for 30 s and extension at 72 °C for 1 min. After 35 cycles, the final extension was rendered at 72 °C for 7 min and the product was allowed to cool at 4 °C.

The PCR product was resolved using a 3.5% agarose gel, which was prepared by dissolving 17.5 g of agarose powder in 500 mL of 1x TAE buffer and heating it in a microwave until it became a clear solution. Ethidium bromide (0.1 mg/mL) was added to the solution after it had cooled to room temperature. The gel was poured into a gel casting tray. The combs were properly positioned to create wells for PCR product loading. After proper solidification, the gel was placed in the electrophoresis tank containing 1x TAE buffer, and the combs were removed from the gel. The PCR product of 10 µL was loaded into individual wells along with a 50 bp ladder (Fermentas, Vilnius, Lithuania). The electrophoresis was run for two to three hours with the power pack adjusted to 5 Volts/cm of run. The gel slabs were observed under a UV trans-illuminator intermittently, and images were recorded in a gel documentation system (Bio-Rad Laboratories Inc., Hercules, CA, USA) after an optimal run of samples. Further, scoring of gels was carried out for each of the SSR marker loci. Based on the corresponding molecular weight of the susceptible (PB1121) and resistant (PR) parents, the allelic pattern displayed by the population was scored as 0 for PB1121, 1 for heterozygotes and 2 for PR allele.

### 4.4. Statistical Analysis

The experimental data from the bioassays were subjected to one-way ANOVA with post hoc Tukey’s honest significant difference test (HSD) at *p* < 0.05 using R base 4.2 for RRKN resistance-related traits, viz., NG, Fm, EM, EEM and MF. The null hypothesis was that there was no significant difference in the means of the variables among the parents and individuals in the F_2:3_ population. The alternative hypothesis (H1) was that there were significant differences in the means of the variables among the different groups in the parents and F_2:3_ population.

A simple linear regression model, Y = α + βX + e [Y = dependent variable (RRKN-resistance trait), X = independent variable (marker genotype), α = Y-intercept of the regression line, β = slope of the line (linear regression coefficient) and e = error variance], was used to test the linkage between the putative markers and the trait variables using R base 4.2. Single-marker analysis was carried out using ICiMapping software version 4.2 for all the RRKN resistance-associated traits [35].

## 5. Conclusions

In conclusion, the present study establishes that PR is a novel source of RRKN resistance in rice. The markers linked to the reported QTLs were mostly ineffective in discerning the resistance conferred by PR. Therefore, we conclude that PR harbours novel genomic regions conferring RRKN resistance. Identification of these novel regions of RRKN resistance will require extensive genome-wide studies so that PR can be used as a potential donor for developing RRKN-resistant cultivars through genomic-assisted breeding.

## Figures and Tables

**Figure 1 plants-13-02271-f001:**
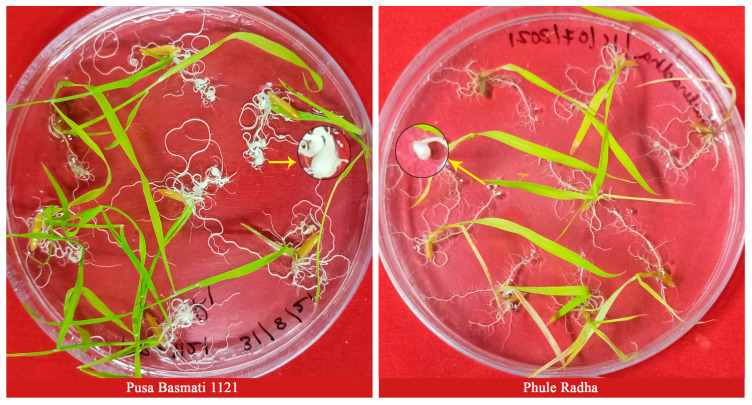
RRKN-infected plantlets of susceptible parent PB1121 and resistant parent PR in Petri dishes with PF-127 medium at 16 dpi (inoculum level-30 J2s per plant). The resistant parent PR showed a reduced number of hook-shaped galls (yellow-coloured arrows) when compared to the susceptible parent PB 1121. The galls have been highlighted by zooming up to 350%.

**Figure 2 plants-13-02271-f002:**
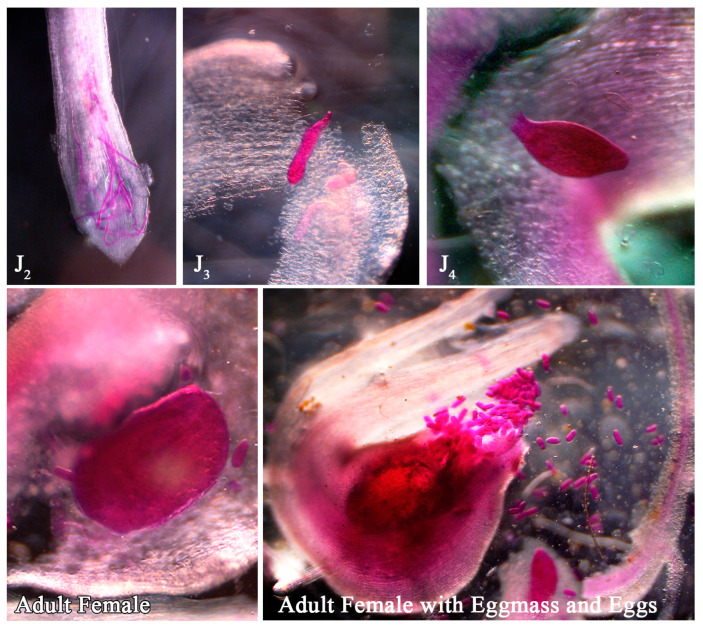
Photomicrographs showing different developmental stages of *M. graminicola* during its life cycle progression in PB1121.

**Figure 3 plants-13-02271-f003:**
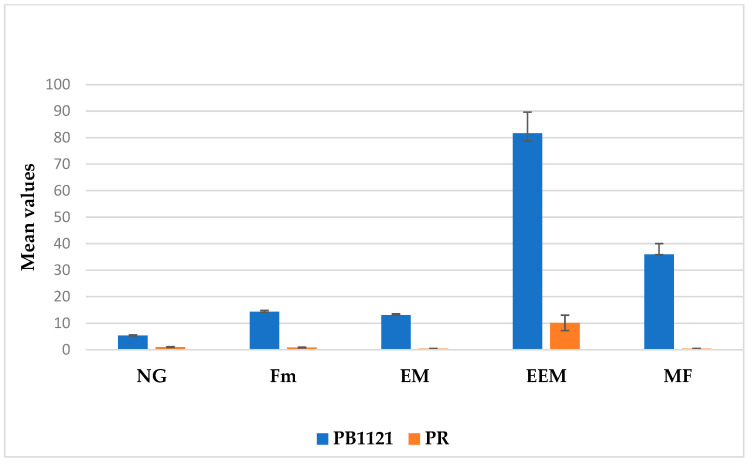
Assessment of RRKN infection in the parents PB1121 and PR in PF-127 media: the resistant parent PR produced significantly lower NG, Fm, EM and EEM, MF compared to the susceptible parent PB1121 (*p* < 0.01, Tukey’s HSD test).

**Figure 4 plants-13-02271-f004:**
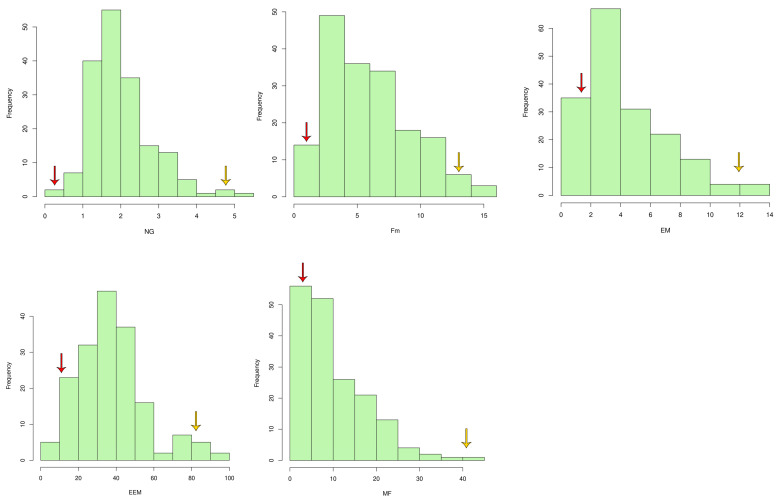
The frequency distribution of RRKN resistance-related traits in the F2:3 population. (Yellow arrow PB1121, Red arrow PR).

**Figure 5 plants-13-02271-f005:**
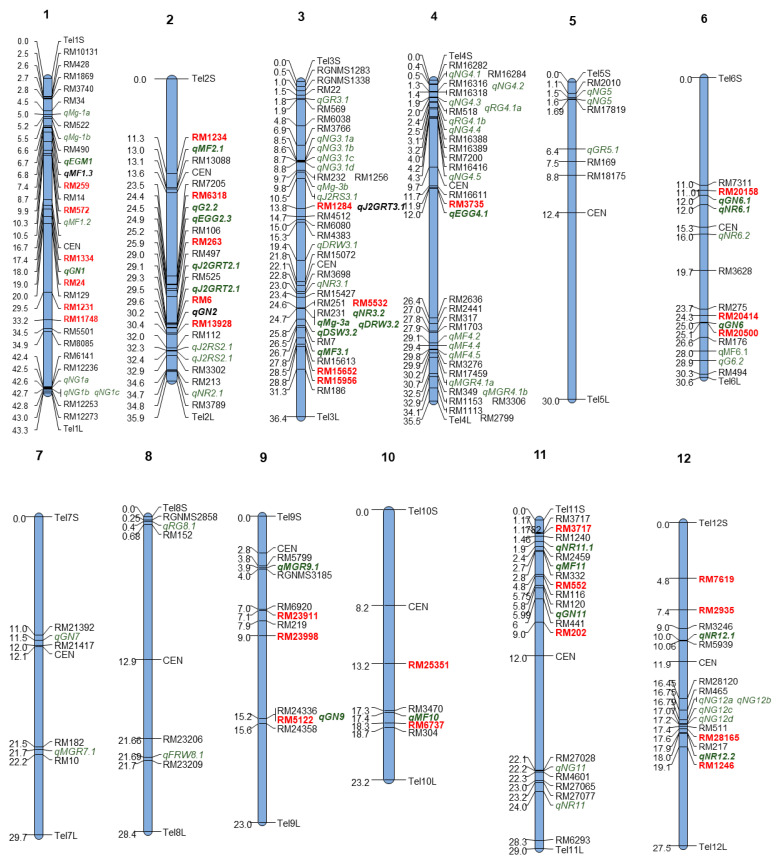
Chromosomal map depicting the physical positions of QTLs reported to be associated with RRKN resistance-related traits along with the markers used in the parental polymorphism survey. The markers that are polymorphic between PR and PB1121 are highlighted in red colour.

**Table 1 plants-13-02271-t001:** One-way ANOVA for RRKN resistance-associated traits in parental lines.

Variable.	DF (Degrees of Freedom)	Sum of Squares	Mean Sum of Squares	F-Value	Pr (>F)
NG	1	274.951	274.951	182.370	0
Fm	1	2634.830	2634.830	675.988	0
EM	1	2313.207	2313.207	613.281	0
EEM	1	74,220.810	74,220.810	74.860	0
MF	1	18,322.230	18,322.230	82.340	0

**Table 2 plants-13-02271-t002:** ANOVA for RRKN resistance traits in F_2:3_ population.

Variables	DF	Sum of Squares	Mean Sum of Squares	F-Value	Pr (>F)
NG	175	2103.85	12.02	9.32	0
Fm	175	31,626.87	180.73	16.26	0
EM	175	23,228.40	132.73	17.56	0
EEM	175	996,565.30	5694.66	8.55	0
MF	175	190,330.60	1087.60	10.25	0

**Table 3 plants-13-02271-t003:** Descriptive statistics of F_2:3_ population for RRKN resistance-related traits.

Traits	Range	Mean	Standard Deviation	Skewness	PB1121	PR
NG	0.17–5.00	2.00	0.78	0.90	5.00	1.00
Fm	0.11–14.33	5.94	3.21	0.53	14.32	0.83
EM	0.00–14.00	4.26	2.74	1.05	13.07	0.43
EEM	0.00–96.60	37.44	17.85	0.76	81.70	10.00
MF	0.00–41.16	9.77	7.66	1.16	35.97	0.40

**Table 4 plants-13-02271-t004:** Associations identified based on simple linear regression (significant associations are presented in bold).

Markers	NG		Fm		EM		EEM		MF	
	R^2^ (%)	*p*-Value	R^2^	*p*-Value	R^2^	*p*-Value	R^2^	*p*-Value	R^2^	*p*-Value
RM24	4	**0.01**	1	0.23	0	0.56	0	0.49	0	0.84
RM1334	3	0.06	1	0.41	0	0.75	0	0.47	0	0.63
RM6	1	0.30	0	0.44	0	0.58	1	0.35	0	0.62
RM13928	0	0.84	0	0.40	0	0.80	0	0.42	0	0.85
RM20158	0	0.94	3	**0.04**	1	0.13	1	0.15	2	0.08
RM20414	0	0.54	1	0.24	1	0.19	0	0.68	1	0.18
RM5122	1	0.13	3	**0.03**	3	**0.02**	5	**0.00**	5	**0.00**
RM552	2	0.11	4	**0.02**	4	**0.02**	1	0.22	4	**0.02**
RM202	1	0.23	0	0.94	0	0.83	0	0.88	0	1.00
RM23911	1	0.13	1	0.19	1	0.21	4	**0.01**	3	**0.02**
RM23998	2	0.07	1	0.14	1	0.12	1	0.30	2	0.06
RM7619	0	0.57	0	0.80	0	0.57	1	0.29	0	0.74
RM2935	0	0.59	0	0.47	1	0.27	0	0.90	0	0.83
RM6293	0	0.39	2	0.11	3	**0.02**	0	0.77	1	0.12
RM1231	0	0.66	0	0.86	0	0.79	0	0.57	0	0.96
RM5532	0	0.73	1	0.35	1	0.30	1	0.12	1	0.21
RM15652	0	0.53	0	0.46	0	0.43	3	**0.03**	2	0.11
RM259	0	0.38	1	0.36	1	0.16	0	0.46	0	0.79
RM28165	0	0.22	0	0.52	0	0.65	1	0.21	1	0.26
RM1246	0	0.28	1	0.36	0	0.41	1	0.18	1	0.16
RM1234	0	0.99	1	0.21	1	0.33	1	0.26	1	0.28
RM6318	0	0.93	0	0.75	0	0.85	0	0.91	0	0.92
RM263	0	0.91	0	0.76	0	0.52	0	0.73	1	0.38
RM15956	0	0.62	1	0.28	0	0.38	1	0.29	1	0.15
RM3735	0	0.70	0	0.70	1	0.18	1	0.13	0	0.58
RM20500	0	0.78	1	0.20	1	0.25	0	0.38	0	0.82
RM25351	0	0.60	0	0.46	0	0.79	2	0.10	0	0.41
RM6737	0	0.99	2	0.11	3	**0.04**	1	0.30	0	**0.03**
RM3717	0	0.38	2	0.15	2	0.11	1	0.39	0	**0.05**

## Data Availability

Summarized data are provided in Supplemental Appendix A.

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
