# Peer review of "Identification of a Promising Novel Genetic Source for Rice Root-Knot Nematode Resistance through Markers Associated with Trait-Specific Quantitative Trait Loci"

_plants, 2024, doi:10.3390/plants13162271_

Round 1
Reviewer 1 Report
Comments and Suggestions for Authors
In this research work, an F2:3 population from the cross PB1121/PR was constructed, and the traits related to RRKN were surveyed. The distribution of traits indicated that a few major effect genes and many minor effect genes may govern the resistance. Markers associated with QTLs reported in the literature were used to test this F2:3 population from the cross PB1121/PR. They can not find a significant association, indicating that the PR may possess novel genomic regions governing resistance to RRKN. However, this work did not provide the QTL mapping results in its F2:3 population from the cross PB1121/PR and can not point out the detailed loci related to RRKN in PR. And so it should add the primary QTL mapping results in F2:3 population before publication.
Author Response
|
Comment 1: In this research work, an F2:3 population from the cross PB1121/PR was constructed, and the traits related to RRKN were surveyed. The distribution of traits indicated that a few major effect genes and many minor effect genes may govern the resistance. Markers associated with QTLs reported in the literature were used to test this F2:3 population from the cross PB1121/PR. They cannot find a significant association, indicating that the PR may possess novel genomic regions governing resistance to RRKN. However, this work did not provide the QTL mapping results in its F2:3 population from the cross PB1121/PR and cannot point out the detailed loci related to RRKN in PR. And so it should add the primary QTL mapping results in F2:3 population before publication. |
Response 1:
Rice root knot nematode is one of the major constraints in adoption of dry-DSR cultivation. It causes significant damage to the crop. There are very few genotypes in the rice germplasm identified to be resistant to RRKN. Many of the reported genotypes are not stable sources of resistance. In this study we have focused on the genotype Phule Radha which has been identified by our group to be highly resistant to RRKN. This genotype has been identified resistant across seven years of rigorous screening using multiple populations of RRKN. This prompted us to conduct assay if Phule Radha carries any of the QTLs reported in other resistance sources. The preliminary mapping was conducted using the markers linked to the already reported QTLs in an F2:3 population. The study led to identification of Phule Radha as resistant source which carry novel genes/QTLs for RRKN resistance.
The present study was to establish the novelty of Phule Radha as a resistance source which do not carry any QTL/gene which has been already reported in other studies.
We have generated RIL population and high-resolution mapping is in progress, which forms a part of another study.
Reviewer 2 Report
Comments and Suggestions for Authors
This manuscript reports gene markers available to evaluate the resistance of the rice cultivar, Phule Dadha. The results will be helpful for researchers, especially nematologists and rice breeders, to develop nematode resistance rice varieties. After revised, this manuscript can be published in the journal.
Title should be changed. First, unless knowing well about rice cultivars, researchers may not understand “Phule Radha” is rice. Second, the authors tried to specify the loci of gene markers that would indicate nematode-resistance in Phule Radha. Thus, the authors should imply in the title that the targeted crop was rice and identified gene markers that represented the rice resistance against the root knot nematode.
English should be edited before the resubmission of the revised manuscript.
L30
Do not italicise the author of the species.
L105-107
Give the statistical values, the F -value with dfs in this case.
K114-119
No need to describe individual numbers indicated in the table. Mention the essence of the results.
Figure 1a and Figure 1b
These should be shown as different figures, like “Figure 1” and “Figure 2”.
Figure 2
Give the label of the vertical axis. Decimals not needed to show.
L146-176
These should be moved to Materials and Methods. The information was not obtained originally in this study but derived from data bases already existing.
L179
The abbreviation for probability, P, should be written in an upper case letter.
L223
Briefly explain what “traditional transplanted rice” is.
L251
Are you sure that major genes are “fewer” than a few?
L288
Refer to the date when the crossing was carried out.
L294
Refer to the date when RRKN was isolated.
L317
Refer to the date when the DNAs were extracted.
L340-345
Specify the variables that were tested in the statistical analyses, in particular stating the hypotheses and models to test.
Comments on the Quality of English LanguageEnglish should be improved before the resumission of a revised manuscript.
Author Response
Point wise response to reviewer's comments has been attached herewith

Round 2
Reviewer 1 Report
Comments and Suggestions for Authors
All the concerns had been clearly clarified in this revised manuscript.
Author Response
Thank you for providing suggestions towards improving the manuscript
Reviewer 2 Report
Comments and Suggestions for Authors
Minor points to collect are as below.
1) L117
This line should be moved above Table 1.
2) Tables 1-3
The df of error term should be referred to.
3) Tables 1 - 4
"F" in these tables should be replaced for, for example "Fm", as the former may be easily confused with F of F-value in ANOVA.
4) Table 4
Are some R-square values expressed in % too small to provide significance for the correlation? For example, that value of RM24 in NG was only 4 with P = 0.01. Check the contribution values.
5. L342
Explain Kharif. If the manuscript is submitted to a domestic journal, no explanations may be needed. However, if submitted to an international journal, a brief explanation should be done. Not all world researchers are popular with the Indian climate.
Comments on the Quality of English Language
English should be checked before resubmission.
Author Response
Comment 1: L117: This line should be moved above Table 1.
Response: Correction incorporated
Comment 2: Tables 1-3: The df of error term should be referred to
Response: Correction incorporated
Comment 3: Tables 1-4: "F" in these tables should be replaced for, for example "Fm", as the former may be easily confused with F of F-value in ANOVA.
Response: “F” has been replaced with “Fm”
Comment 4: Table 4: Are some R-square values expressed in % too small to provide significance for the correlation? For example, that value of RM24 in NG was only 4 with P = 0.01. Check the contribution values.
Response: We have re-checked the data and analysis part and we confirm that there is no error in the reported results. Lower contribution values (R2) may also lead to significant associations. However, these associations may be false positives, for which additional analysis is required. In our case, upon analysis for single marker analysis, the LOD value for this marker was below the threshold. Therefore, we declared this marker is not associated with the trait NG.
Comment 5: L342: Explain Kharif. If the manuscript is submitted to a domestic journal, no explanations may be needed. However, if submitted to an international journal, a brief explanation should be done. Not all world researchers are popular with the Indian climate.
Response: Kharif season refers to wet season. Therefore, we have changed from Kharif to wet season although the manuscript.